# Present and Future of Immunotherapy for Triple-Negative Breast Cancer

**DOI:** 10.3390/cancers16193250

**Published:** 2024-09-24

**Authors:** Sushmitha Sriramulu, Shivani Thoidingjam, Corey Speers, Shyam Nyati

**Affiliations:** 1Department of Radiation Oncology, Henry Ford Cancer Institute, Henry Ford Health, Detroit, MI 48202, USA; 2Department of Radiation Oncology, University of Michigan, Ann Arbor, MI 48109, USA; 3Department of Radiation Oncology, UH Seidman Cancer Center, University Hospitals Case Medical Center, Case Western Reserve University, Cleveland, OH 44106, USA; 4Henry Ford Health + Michigan State University Health Sciences, Detroit, MI 48202, USA; 5Department of Radiology, Michigan State University, East Lansing, MI 48824, USA

**Keywords:** TNBC, triple-negative breast cancer, immunotherapy, immune checkpoint inhibitors, radiation therapy, chemotherapy

## Abstract

**Simple Summary:**

Immunotherapy has changed the way we treat triple-negative breast cancer (TNBC), a type of breast cancer that does not have common markers like estrogen, progesterone, or HER2. TNBC has the worst outlook among breast cancers, but it may respond better to immunotherapy because it often shows higher levels of PD-L1 and has more immune cells attacking the tumor. Recently, the FDA approved pembrolizumab (Keytruda) combined with chemotherapy for advanced TNBC, offering new hope for patients. However, not all patients respond equally well, mainly because not everyone has high PD-L1 levels. To improve treatment success, combining immunotherapy with other treatments like chemotherapy, targeted therapies, or radiation seems promising. This review explains TNBC and immunotherapy, discusses current and future combination treatment strategies, and explores the challenges and potential new approaches that might soon be available for treating TNBC.

**Abstract:**

Triple-negative breast cancer (TNBC) lacks the expression of estrogen receptors (ERs), human epidermal growth factor receptor 2 (HER2), and progesterone receptors (PRs). TNBC has the poorest prognosis among breast cancer subtypes and is more likely to respond to immunotherapy due to its higher expression of PD-L1 and a greater percentage of tumor-infiltrating lymphocytes. Immunotherapy has revolutionized TNBC treatment, especially with the FDA’s approval of pembrolizumab (Keytruda) combined with chemotherapy for advanced cases, opening new avenues for treating this deadly disease. Although immunotherapy can significantly improve patient outcomes in a subset of patients, achieving the desired response rate for all remains an unmet clinical goal. Strategies that enhance responses to immune checkpoint blockade, including combining immunotherapy with chemotherapy, molecularly targeted therapy, or radiotherapy, may improve response rates and clinical outcomes. In this review, we provide a short background on TNBC and immunotherapy and explore the different types of immunotherapy strategies that are currently being evaluated in TNBC. Additionally, we review why combination strategies may be beneficial, provide an overview of the combination strategies, and discuss the novel immunotherapeutic opportunities that may be approved in the near future for TNBC.

## 1. Introduction

Triple-negative breast cancer (TNBC) represents 15–20% of breast cancers (BCs) that are newly diagnosed and is a subtype with the fewest approved targeted therapies [1]. TNBC does not overexpress human epidermal growth factor receptor 2 (HER2) and lacks expression of either estrogen receptor (ER) or progesterone receptor (PR) [2]. Clinically, TNBC tumors are typically larger upon diagnosis and tend to develop nodal metastasis to the draining lymphatics [3]. This subtype is known for developing more lethal metastases which are more likely to originate in viscera, especially the brain and lungs, and less likely to spread to the bone [4]. TNBC patients are at a greater risk of early relapse compared to those with other BC subtypes, and only a subset of TNBC patients show a more favorable response to chemotherapy [4,5]. TNBC recurrence often peaks between the first- and third-year following diagnosis, then sharply declines in the years that follow. Relapses after eight or ten years are extremely rare [5]. Developments in treatment strategies remained limited for years, and cytotoxic chemotherapy continues to be the primary systemic treatment for TNBC and is often used in the neoadjuvant setting [1]. This enables a reduction in tumor burden and facilitates an in vivo evaluation of treatment response, making pathological complete response (pCR) a valuable prognostic marker for survival [6]. While there are several drugs targeting ER and HER2 in clinical subgroups, the paucity of progress in developing targeted therapies for TNBC is apparent [1]. The understanding of the intricate molecular and genetic basis for TNBC has steadily improved over the last decade or so, with several classification schemas for TNBC proposed [3,7,8,9]. Using emerging technologies, such as next-generation sequencing (NGS), Lehmann et al. validated both intratumoral and intertumoral heterogeneity and simplified the molecular classification of TNBC into six different subtypes: basal-like 1 and 2 (BL1 and BL2), immunomodulatory (IM), luminal androgen receptor (LAR), mesenchymal (M), and mesenchymal stem-like (MSL) [10]. In 2016, Lehmann revised the classification into four distinct subtypes: BL1, BL2, M, and LAR. IM and MSL were left out due to their low cellularity and dependability on tumor-infiltrating lymphocytes (TILs) and tumor-associated stromal cells [11].

Despite the widely held notion that BC is not immunogenic, numerous studies demonstrated that TNBC can activate the immune system. A subset of TNBC has BRCA 1/2 mutations [12], which are associated with higher tumor immunogenicity [13]. However, the immunogenicity of TNBC goes beyond BRCA1/2 mutations since the TNBC subtype has the greatest incidence in patients with a robust tumor immune infiltrate. Moreover, TNBC prognostic signatures include B cells which can secrete antibodies that bind to tumor antigens and amplify the adaptive immune responses [14]. Compared to other BC subtypes, TNBC displays lower clonal heterogeneity and higher immune gene expression [15]. The immune microenvironment of TNBC is a dynamic, intricate network of different immune cell populations, cytokines, and signaling pathways [16]. The immunological components of the TNBC tumor microenvironment (TME) are complex, as described in Figure 1.

Significant recent advancements have led to FDA approval of several drugs for TNBC (Table 1). Novel strategies such as immunotherapy, ionizing radiation therapy [17], platinum agents [18], and PARP inhibitors [19] have been explored to increase the pCR rates. TNBC is most likely to benefit from immunotherapy because several studies reported higher tumor mutational burden, increased expression of programmed cell death-ligand1 (PD-L1), and higher TILs in the TME compared to the other BC subtypes [20,21]. Based on several clinical trials, the immunotherapy and chemotherapy combination was approved in both early and advanced TNBC settings [22]. In this review, we discuss the different types of immunotherapy strategies that are currently employed in TNBC, articulate why combination strategies may be beneficial, provide an overview of the combination strategies, review the current clinical challenges encountered, and provide insight into potential future clinical developments. 

## 2. Current Immunotherapy Approaches: Approved Treatments and Agents under Investigation for TNBC

The treatment landscape for TNBC has evolved significantly in recent years, with immunotherapy emerging as a promising avenue. This section provides an overview of the current clinical immunotherapy approaches being investigated for TNBC. These include cytokine therapies, monoclonal antibodies (mAbs), antibody–drug conjugates (ADCs), immune checkpoint inhibitors (ICIs), cancer vaccines, adoptive cell therapies (ACTs), and oncolytic virus therapy (OV). Each of these strategies offers a unique mechanism to boost antitumor immunity, with several approaches showing potential in clinical trials. The following illustration (Figure 2) summarizes these diverse therapeutic strategies.

### 2.1. Cytokines

Cytokines are small proteins that are essential for immune response and cell signaling [23]. While cytokine-based therapies have been explored in several cancers including BC, their role in TNBC treatment is still actively explored. Cytokines regulate the host immune response to cancer cells and aid in prompting cancer cell death, making cytokine-based immunotherapy an intriguing possibility in cancer treatment [24]. The two cytokines approved by the FDA for the treatment of cancer, though not breast cancer, are IL-2 and IFN-α; nevertheless, their high toxicity profile has limited their use [23]. Several other cytokines including GM-CSF [25], IL-12, IL-15, and TNF are being tested in numerous clinical trials for their safety and effectiveness as cancer treatments [26]. Moreover, as single-agent immunotherapies, over 40 known cytokines have been approved for a restricted range of indications, including the treatment of cancer [27]. A recombinant adenovirus expressing IL-12 (AdIL-12) administered intratumorally has been demonstrated to cause substantial tumor regression in animal models of BC [27]. Patients with metastatic TNBC (mTNBC) demonstrated improved antigen presentation and a treatment-related spike in CD8+ TIL density following intratumoral administration of IL-12 (phase-1 pilot study) [28]. An engineered cytokine called empegaldesleukin or NKTR-214 preferentially activates the IL-2 receptor with a focus on metastatic solid tumors, including TNBC (phase-1 trial; NCT02869295) [29]. Cytokine activity facilitates both tumor-promoting and tumor-suppressive effects. Proinflammatory cytokines such as TNF-α and IL-6, for instance, regulate immunological interactions to promote anticancer effects. Cytokines present in the TME promote angiogenesis, epithelial-to-mesenchymal transition, invasion, and tumor growth—all processes linked to cancer development [30]. Several factors contribute to the poor efficacy of cytokine immune therapy including short half-life, increased toxicity, and low efficacy. High intratumoral cytokine dosages might cause systemic adverse effects including renal insufficiency, hypotension, neuropsychiatric symptoms, and respiratory failure [31]. Similarly, patients also have difficulty tolerating systemic treatment with recombinant IFN-α [32]. IRX-2, a novel therapy comprising numerous cytokines, demonstrated activation of the TME by elevating TIL numbers, PD-L1 expression, and lymphocyte activation in early TNBC patients in a phase-I study. A phase-II follow-up trial is underway (NCT04373031) [33]. Even though cytokines were thought to have several benefits when used as a monotherapy, most clinical trials that employed systemic cytokine monotherapy failed. This could be due to inadequate cytokine concentrations in the tumor upon parenteral administration, as well as major toxicities related to the activation of humoral or cellular checkpoints [34].

### 2.2. Monoclonal Antibodies

The goal of using humanized mAbs is to decrease immunotolerance and boost the immune response against tumors by blocking immunosuppressive checkpoints that the tumor uses to evade immune system control [35]. Trastuzumab was the first FDA-approved monoclonal antibody for the treatment of HER2-positive breast cancer [36]. TNBC tumors do not express HER2; thus, these patients cannot be treated with trastuzumab or other HER2-specific agents [37]. In 2009, the FDA authorized bevacizumab (Avastin), a humanized mAb that specifically targets VEGF-A. VEGF regulates tumor-induced immunosuppression in addition to blood vessel development [38]. Therefore, bevacizumab’s immunomodulatory characteristics open possibilities for novel combination treatment approaches [38]. In 2010, the approval of bevacizumab for breast cancer was recommended for withdrawal by the Office of New Drugs (OND). Significant benefits were not confirmed by the required follow-up trials, and increased serious adverse events with a lack of survival benefit were revealed, leading to the conclusion that the risks outweigh the benefits of this indication [39]. Aspartic protease Cath-D is an extracellular target unique to TNBC. As a promising immunotherapy, an immunomodulatory antibody-based approach against Cath-D is presently in its developmental phase to treat TNBC patients [40].

### 2.3. Antibody–Drug Conjugates

ADCs can identify antigens that are tumor-specific or overexpressed in tumors and thus can kill cancer cells via antigen-dependent cell-mediated cytotoxicity (ADCC) [41]. ADCs utilize the specificity of mAbs on cellular-antigen identification to administer potent cytotoxic drugs in a tailored approach [42]. TNF receptor superfamily members such as CD40 can help DCs to stimulate antitumor T cells and retrain macrophages to kill tumor stroma. The efficacy of anti-CD40 antibodies has been evaluated in several clinical trials [43].

ADC trastuzumab deruxtecan (DS-8201) comprises a cytotoxic topoisomerase I inhibitor and an anti-HER2 antibody connected by a cleavable tetrapeptide linker [44]. Its capacity to produce cytotoxic action against antigen-negative cells (bystander effect) may have led the FDA to approve it in BC patients who have been pretreated with trastuzumab emtansine (TDM1) [45]. The DESTINY-Breast04 and DESTINY-Breast06 trials have yielded critical data on the efficacy of trastuzumab deruxtecan in treating low-HER2 breast cancer. DESTINY-Breast04 demonstrated that patients with low-HER2 breast cancer (IHC 1+ or IHC 2+ without FISH amplification) had significantly superior progression-free survival (PFS) when treated with trastuzumab deruxtecan compared to those who received the physician’s choice of cytotoxic chemotherapy. This trial established trastuzumab deruxtecan as a viable treatment option for low-HER2 breast cancer, highlighting its potential to change clinical practice [46].

Similarly, the DESTINY-Breast06 trial expanded the understanding of trastuzumab deruxtecan’s efficacy by showing benefits for patients with “ultra-low”-IHC breast cancer (IHC 0 but with subtle HER2 expression). This finding is particularly notable as it extends the applicability of trastuzumab deruxtecan to a broader patient population, including those previously not considered for HER2-targeted therapies (NCT04494425). These results are especially significant for TNBC patients, who traditionally have limited treatment options. Moreover, it is important to underscore the CNS activity of trastuzumab deruxtecan, as its ability to penetrate the central nervous system could provide substantial benefits for patients with brain metastases, a common and challenging complication in breast cancer. The TUXEDO-1 trial (NCT04752059) demonstrated that trastuzumab deruxtecan achieved a high intracranial response rate in patients with active brain metastases from HER2-positive breast cancer, establishing it as a viable treatment option for this condition [47]. Another ADC, Disitamab vedotin, consists of a HER2 mAb conjugated to cytotoxic drug monomethyl auristatin E with a cleavable linker and is also being tested in clinical trials for several solid tumors including low-/HER2+ BCs [48].

Sacituzumab govitecan is the first ADC to receive FDA approval to treat TNBC which consists of a human trophoblast cell-surface antigen-2 (TROP2) antibody that is connected with a topoisomerase I inhibitor (SN-38) via a unique hydrolyzable linker [49]. Recently, a phase-III trial (ASCENT) demonstrated noticeably prolonged OS and PFS while using sacituzumab govitecan instead of single-agent chemotherapy [50]. Apart from HER2 and TROP2-based ADC, the activities of folate receptor alpha (FRα) and zinc transporter LIV-1-based ADC have been clinically evaluated for TNBC [40]. Ladiratuzumab vedotin is an ADC that targets Syndecan-1 on cancer cells. It binds to these cells, is internalized, and releases a cytotoxic agent that disrupts microtubules, leading to cell cycle arrest and apoptosis [51]. In early-phase clinical trials, ladiratuzumab vedotin is being explored as a monotherapy for patients with BC; some of these studies have already shown encouraging results [51]. In a phase-I, multi-part, dose-escalation SGNLVA-001 trial, mTNBC exhibited a superior overall response rate (ORR) and disease control rate (DCR) with Ladiratuzumab vedotin [52]. Recently, Tsai and colleagues reported an ORR of 28% with Ladiratuzumab vedotin at 1.25 mg/kg, indicating the favorable activity of the ADC [53].

Recently, a newly developed ADC (ESG401), which consists of a humanized anti-TROP2 IgG1 monoclonal antibody connected to the Topoisomerase I Inhibitor SN-38 through a stable cleavable linker, demonstrated promising effectiveness and tolerability in a phase-Ia trial [54]. Moreover, the effectiveness of ESG401 in treating brain metastases in first-line mTNBC patients corresponds with our prior findings in late-line mTNBC and HR+/HER2− BC patients (NCT04892342) [55]. Datopotamab deruxtecan (Dato-DXd) is another ADC where the antibody datopotamab, targeting TROP2 on breast cancer cells, is linked to the cytotoxic drug DXd. Once datopotamab binds to TROP2 and is internalized, the linker breaks down, releasing DXd to kill the cancer cell [56]. Dato-DXd’s ability to recruit immune cells to cancer sites suggests that combining it with durvalumab, which blocks PD-L1 and enhances immune cell activity, may enhance its effectiveness. In the phase-I study of Dato-DXd (NCT03401385), promising antitumor activity and a manageable safety profile were observed in patients with heavily pretreated advanced HR+/HER2− breast cancer and TNBC [57]. The ongoing TROPION-Breast03 trial (NCT05629585) will compare Dato-DXd alone or with durvalumab against standard care in patients with non-mTNBC with residual cancer cells post-surgery [56]. Thus, ADCs continue to play an ever evolving and significant role in the management of mTNBC.

### 2.4. Immune Checkpoint Inhibitors

ICIs kill tumor cells by disabling the immune system’s “braking” function on the immune cells that attack cancer. ICIs target the three best-characterized targets, PD-L1 (also called as B7 homolog 1) and PD-1 (also known as CD279), while blocking cytotoxic T-lymphocyte-associated protein 4 (CTLA-4/CD152) [58]. The mechanism of action of ICIs targeting PD-L1, PD-1, and CTLA-4 is shown in Figure 3. ICIs have the potential to be beneficial for both immunoinflammatory and immunological-suppressive types, potentially changing the TME from an “immune cold” to an “immune hot” phenotype [59]. The immunosuppressive PD-1 protein is mostly expressed on the cell surface (i.e., plasma membrane) of B, T, myeloid, and NK cells of the immune system [60]. The FDA-approved PD-1/PD-L1 inhibitors include atezolizumab, avelumab, cemiplimab, durvalumab, nivolumab, and pembrolizumab [61]. PD-L1 expression is directly correlated to histological grade and lymphocyte infiltration and is observed in 20–30% of TNBC cases [61].

Pembrolizumab was found to be safe with a good ORR in TNBC patients (KEYNOTE-012; [62]) and pembrolizumab monotherapy provided long-lasting antitumor efficacy in patients with both early and advanced PD-L1-positive TNBC with a combined positive score ≥ 1 [63,64]. PD-1 inhibitor JS001 demonstrated good safety and efficacy in mTNBC patients (NCT02838823) who failed prior multi-line treatments [65,66]. Pembrolizumab received approval based on the KEYNOTE-355 trial (NCT02819518), a multicenter, double-blind, randomized, placebo-controlled study involving patients with locally recurrent unresectable or metastatic TNBC who had not previously received chemotherapy for metastatic disease [66]. Similarly, atezolizumab monotherapy offered enduring clinical advantages for patients with mTNBC (NCT01375842; [67]), while avelumab provided an ORR of 44.4% (PD-L1 ≥ 10%) and 2.6% (PD-L1 < 10%) in TNBC patients (JAVELIN trial; [68]). The FDA initially approved atezolizumab in combination with nab-paclitaxel for first-line treatment of TNBC based on the IMpassion130 trial. However, this approval was later withdrawn following the negative results of the IMpassion131 trial [69].

T cells receive positive and negative feedback from the CD28 and CTLA-4 receptors, respectively [70]. CTLA-4 maintains T-cell homeostasis since it specifically regulates CD4+ T-cell responses [71]. Importantly, tissues from lymph node metastases show considerably higher CTLA-4 levels than tissue samples from the original breast tumor as seen in axial lymph node (ALN) metastasis [72]. There is presently no approved CTLA-4 inhibitor that can be used exclusively for TNBC, but ipilimumab is FDA-approved to treat several other cancers.

### 2.5. Vaccines

Known antigens associated with breast tumors are the main target of therapeutic vaccines, which work by actively immunizing against the tumor. A patient-specific tumor mutanome is used in cutting-edge settings to produce vaccines [73]. Vaccines can modify the TME through chemokines which can directly impact tumor growth as well as cytotoxic CD8+ T-cell (CTL) and NK responses. Several independent approaches have been used to develop therapeutic vaccines that use DC, DNA, RNA, peptides, carbohydrates, or all of the above [74]. CD4+ helper T lymphocytes can be stimulated by AE37, which is an Ii-Key hybrid of the MHC class II peptide. The randomized phase-II trial comparing GM-CSF alone with the AE37 + GM-CSF vaccine to prevent BC recurrence revealed no statistically significant differences in the five-year DFS between the treatment arms [75,76]. Similarly, a phase-1 trial with FRα peptide vaccine was well tolerated and produced responses that lasted over a year in more than 90% of patients with BC [77]. Poxvirus used in the PANVAC vaccine encodes mucin-1 (MUC-1) and carcinoembryonic antigen (CEA) along with T-cell-stimulating proteins LFA-350, ICAM-1, and B7.1 [78]. Favorable clinical responses to this vaccine have been observed in a limited number of patients [79]. Similarly, mixed subtypes of BC subjects treated with an autologous DC vaccine pulsed with various p53 peptides resulted in stable disease in ~30% of patients, and a small subgroup of these had increased CD8+ T-cell responses [80]. Autologous DC vaccine increased PFS to over 3 years in a subgroup of ER/PR double-negative [81] and PR-negative stage IV BC subjects [82]. Early clinical investigations of the DC vaccine including hTERT [83] peptides and FRα [77] have demonstrated T-cell activation, further supporting the role of the DC vaccine in BC management. These include a novel alpha-lactalbumin vaccine in patients with stage II-III TNBC (phase I) and in individuals at risk for TNBC who are planning to undergo preventative bilateral mastectomy (phase I) [84], STEMVAC, a DNA plasmid-based vaccine (phase II, NCT05455658) on patients with curatively treated stage I–III TNBC. STEMVAC targets proteins expressed on breast cancer stem cells, working to enhance the immune system’s ability to detect and eliminate the cancer cells responsible for the disease [85].

### 2.6. Adoptive Cell Therapy

T cells are crucial for cell-mediated immunity. Two forms of ACT that can alter natural T cells ex vivo and reintroduce them into the body to make them more potent tumor-destroying agents are chimeric antigen receptor (CAR) T-cell and T-cell receptor (TCR)-engineered T-cell therapies [86]. CAR T cells are designed to exclusively identify surface antigens by fusing antibody fragments on the T cell’s antigen-binding region. In contrast, MHC-expressed intracellular antigens are recognized by TCRs through the utilization of an alpha–beta chain heterodimer. Consequently, since TCRs may target a larger variety of antigens than CAR-T, they might be more advantageous in solid tumors [86]. Enhancing CAR-T cell infiltration in tumor tissues is a main obstacle for CAR-T therapy in BC. This obstacle may be addressed by combining the delivery of CAR-T cells with strong stimulation of antigen-presenting cells (APCs) to produce chemotactic cytokines [87]. Receptor tyrosine kinase c-Met is overexpressed in approximately 50% of BCs. The intratumoral injections of c-Met-CAR-T cells were well tolerated and induced an inflammatory response in metastatic BC patients with c-Met-expression in a phase-0 trial (NCT01837602) [88]. Mesothelin is overexpressed in TNBC, which is linked to a poorer prognosis [89]. This led to the development of mesothelin-specific CAR-T cells. Initial findings from a phase-I/II trial (NCT02414269) in patients with advanced solid tumors demonstrated antitumor activity of mesothelin-targeted CAR-T cells without any significant toxicities [90]. Similarly, a combination of mesothelin-targeted CAR-T cells with pembrolizumab was found to be safe and well tolerated in patients with malignant pleural disease (NCT02792114) and demonstrated antitumor efficacy [91]. Over 90% of TNBC cases express high MUC1 protein, which is linked to a poor prognosis [92]. An anti-MUC1 CAR-T cell-based phase-I clinical trial is currently underway (NCT04020575) for patients with advanced MUC1-positive BC [93]. Another well-known marker for BC’s adverse prognosis is CEA [94]. A phase-I trial that aimed to analyze the safety and tolerability of anti-CEA T-cell therapy (NCT00673829) in patients with metastatic BC has been suspended without any published results [95]. Luen et al. showcased how TILs could be a crucial factor in adapting clinical trial designs. Currently, TILs do not have clinical utility in any cancer type, and thus should not be utilized as a biomarker to tailor clinical therapies in daily practice. To fully investigate their clinical relevance, the next logical progression would be to consider using TILs as an adaptive factor in clinical trials [96]. Additional clinical testing of adoptive cell therapies in TNBC is warranted to improve the clinical outcome in subjects with TNBC, especially late-stage and mTNBC.

### 2.7. Oncolytic Virus Therapy

Natural or genetically engineered viruses that can proliferate selectively in cancer cells without harming healthy cells are known as OVs [97,98]. OVs may lyse tumor cells by infecting them directly, multiplying within them, or stimulating the immune system [99]. OVs are designed to target several stages in the cancer–immunity cycle. Oncolytic viruses cause immunogenic cell death, which triggers the innate and adaptive immune systems by releasing danger signals and neo-antigens [100]. At present, the only type of OV authorized for cancer therapy is talimogene laherparepvec (T-VEC), a herpes simplex virus-1 (HSV) modified to express GM-CSF [101]. An additional phase-II trial (NCT02658812) assessed the effectiveness of intratumoral T-VEC as monotherapy for locoregionally inoperable BC recurrence, irrespective of whether there was a distant recurrence. The results demonstrated that uncontrolled disease development made intratumoral T-VEC monotherapy less effective, and concurrent systemic treatment administration may be necessary [102]. The most researched OV for BC management is adenovirus. The ICOVIR-7 trial included patients with advanced and refractory solid cancer including BC. Although the OV was declared safe and a majority of subjects (16 out of 18) developed neutralizing antibodies, all BC subjects (*n* = 3) failed to reach efficacy endpoints [103]. Conversely, Ad5/3-D24-GM-CSF, an OV that codes for GM-CSF, effectively immunized patients with advanced BCs including TNBC [104]. Currently, an OV, MEM-288, that carries recombinant chimeric CD40 (MEM40) and human interferon beta (IFNβ) is being investigated (NCT05076760) in various solid cancers including TNBC [105]. In vivo and ex vivo testing of several different OVs, including Coxsackie, Maraba, measles, Newcastle disease, Polio, and Vaccinia, has cleared the path for human safety studies [106]. A phase-I clinical trial (NCT01846091) using a measles virus that encodes human thyroidal sodium iodide symporter (MV-NIS) is presently being evaluated in a range of cancers including mTNBC [107].

## 3. Rationale of Combining Immunotherapy with Other Therapies

TNBC is an aggressive disease and often develops resistance to standard-of-care (SOC) treatment. Thus, immunotherapy in combination with SOC is expected to improve the outcome for several reasons: (1) Different therapies target cancer cells through distinct mechanisms. Combining immunotherapy with chemotherapy, targeted therapy, or radiation therapy can attack cancer cells through multiple pathways simultaneously, resulting in a robust response [108]. (2) Some therapies can increase the immune system’s capacity to identify and target cancer cells. For example, chemotherapy can induce immunogenic cell death, releasing tumor antigens that activate the immune response, thereby synergizing with immunotherapy [109]. (3) TNBC often develops resistance to single-agent therapies [110]. Combination therapies can target several pathways implicated in tumorigenesis and immune evasion, reducing the likelihood of developing resistance [111]. (4) Not all patients respond to immunotherapy alone. Combining immunotherapy with other therapies may broaden the spectrum of patients who benefit from treatment, improving overall outcomes [112]. (5) Combining lower doses of different therapies may reduce individual treatment-related toxicities while maintaining efficacy, improving patients’ quality of life [111]. (6) Targeting TNBC with a combination of therapies can potentially decrease the risk of metastasis or recurrence by eradicating residual cancer cells that may metastasize to other tissues/organs [113]. Overall, combination strategies with immunotherapy represent a comprehensive approach to treating TNBC, addressing its heterogeneity and resistance mechanisms while maximizing therapeutic efficacy and minimizing toxicity. Several key TNBC clinical trials that include(d) immunotherapy in combination with other treatment modalities are listed in Table 2 (non-exhaustive list).

### 3.1. Combination of Immunotherapy with PARP Inhibitors

The combination of Poly ADP-ribose polymerase (PARP) inhibitors (PARPis) with immunotherapy is an active area of investigation for TNBC therapeutics. PARP plays an important role in the DNA repair process. Olaparib and talazoparib are key PARPis approved to treat BRCA1/2 mutant cancers, while niraparib is specifically approved for use in gynecologic cancers [114]. PARP inhibitors were approved by the FDA for patients with metastatic and early TNBC who have germline mutations in BRCA1/2 [115,116]. However, recent studies have shown that PARPis may also have immunomodulatory effects that could enhance the efficacy of immunotherapy in TNBC [117]. TNBCs with BRCA mutations or other DNA repair deficiencies are significantly more sensitive to PARPis [116]. Several clinical trials are currently investigating the combination of PARPis with immunotherapy in TNBC.

Patients with germline BRCA1/2 mutation-associated HER2-negative BC undergoing treatment in the first- through third-line of therapy were randomized to receive either olaparib or chemotherapy in the phase-3 OLYMPIAD trial [115]. Olaparib did not considerably increase OS in the study population as compared to chemotherapy, but the median PFS was 2.8 months longer. However, among patients receiving first-line treatment, olaparib increased OS by 7.9 months compared to chemotherapy alone [116]. This led to FDA approval for Olaparib therapy in women with TNBC with germline BRCA mutations. In the OlympiA trial (NCT02032823), adjuvant olaparib significantly enhanced invasive and distant disease-free survival in patients with high-risk, HER2-negative early breast cancer and germline BRCA1 or BRCA2 mutations, compared to a placebo. However, olaparib had a minimal effect on the overall quality of life reported by patients [118]. Likewise, patients receiving first- through fourth-line therapy for germline BRCA1/2 mutation-associated HER2-negative BC in the phase-3 EMBRACA trial were randomly assigned to receive talazoparib or chemotherapy [115]. In comparison to chemotherapy, talazoparib had a greater ORR and mPFS, but talazoparib did not considerably increase OS when compared to chemotherapy [115]. Based on these results, PARPi monotherapy was suggested as the SOC for metastatic HER2-negative BC patients with BRCA1/2 mutations [119]. Despite having strong response rates, PARPi-induced responses are generally less durable. Although checkpoint inhibitor monotherapy has a lower response rate than PARPis, it produces longer-lasting effects in mTNBC [120].

Comparing the results of the phase-II KEYNOTE-162 trial [121], which combined PD-1 checkpoint inhibitor pembrolizumab with PARPi niraparib, to trials with PARPi monotherapy, it became evident that patients with BRCA1 or BRCA2 pathogenic variants (PVs) displayed long-lasting responses with combination treatment [122]. In the phase-I/II MEDIOLA basket trial combining olaparib and durvalumab, the patients with metastatic HER-2-negative BC with germline BRCA1/2 PVs were explicitly included in one of the four cohorts [123]. Overall, the combined regimen was well tolerated but it is unclear whether the combined approach would be more effective in this group of patients with germline BRCA1/2 PVs than the PARP inhibitor alone, especially in terms of extending the duration of response [123]. Avelumab in combination with talazoparib has been assessed in two JAVELIN basket trials for patients with previously treated solid malignancies, including BC. The JAVELIN trials showed an acceptable level of safety [124]. While initial results from early-phase clinical trials have shown promising activity, larger randomized controlled trials are needed to establish the optimal combination regimen, patient selection criteria, and long-term outcomes. Combining PARP inhibitors with immunotherapy may also pose challenges, such as increased toxicity or the development of resistance. Therefore, careful monitoring and management of adverse events are essential moving forward.

### 3.2. Combination of Immunotherapy with Chemotherapy

Based on several preclinical and clinical studies, it is clear that several chemotherapies kill tumor cells by promoting the recruitment and maturation of APCs, enhancing the antigen presentation, and encouraging T-cell activation [125]. Additionally, by releasing MHC molecules and cell surface antigens, chemotherapy drugs improve the immunogenicity of tumors [126]. Chemotherapy-induced transient immunosuppression results in an enormous release of cyto- and chemokines which boosts immune cell infiltration and activation. Therefore, the combination of PD-(L)1 inhibitors and chemotherapy is a viable strategy to improve immunotherapy efficacy and promote synergistic antitumor activity [21]. Numerous clinical trials combining immunotherapy and chemotherapy are being carried out based on this concept with significant clinical advantage attained.

Most clinical trials currently use immunotherapy and chemotherapy together, in part because the response to ICIs is slower, whereas chemotherapeutic agents kill tumor cells and alter the TME during the therapy period [21]. Previously untreated advanced TNBC patients were treated with pembrolizumab plus chemotherapy or placebo plus chemotherapy as the first-line treatment in KEYNOTE-355. The chemotherapy regimens were chosen by the treating physicians and included gemcitabine/carboplatin, paclitaxel, and nab-paclitaxel [127]. Patients treated with the pembrolizumab and chemotherapy combination had a significant and clinically meaningful increase in PFS compared to the placebo–chemotherapy group with a combined positive score (CPS) ≥ 10. CPS is determined by the ratio of PD-L1-positive cells (tumor cells, lymphocytes, and macrophages) to the total number of tumor cells using the 22C3 assay [128]. The release of follow-up data showed that when pembrolizumab was added to chemotherapy, OS increased by about 7 months in the CPS ≥ 10 group and side effects were tolerable [128]. In the phase-IB/II KEYNOTE-150 trial, patients with mTNBC who had undergone at least two lines of previous therapy were treated with pembrolizumab plus eribulin mesylate. Of the 167 patients recruited, 40% were categorized as stratum 1 since they had not previously received systemic therapy. In this group of patients, the survival was greatest for PD-L1-positive individuals [129]. The IMpassion130 phase-III trial [66] further validated the efficacy of this immuno-chemotherapy combination for mTNBC patients who did not receive systemic therapy, following a phase-Ib trial (NCT01633970) that showed the safety of atezolizumab plus nab-paclitaxel in patients with locally recurrent or mTNBC [130]. Based on preliminary findings, adding atezolizumab was associated with a benefit for PFS in both PD-L1-positive and intention-to-treat (ITT) populations. Atezolizumab was found to significantly enhance OS in the PD-L1-positive group from 18.0 months to 25.0 months in the second set of intermediate findings; however, in the ITT population, there was no significant difference [131]. As a result, the FDA approved atezolizumab in March 2019 for use as a first-line treatment for patients with late-stage TNBC in combination with nab-paclitaxel [132]. The combination of AKT inhibitor ipatasertib with atezolizumab + paclitaxel/nab-paclitaxel demonstrated excellent efficacy in treating advanced and mTNBC [133].

Nevertheless, these different results were noted in the phase-III IMpassion131 trial which examined atezolizumab plus paclitaxel as first-line therapy for patients with advanced or mTNBC and compared the results to placebo plus paclitaxel. This study did not find any discernible differences in PFS between the two groups based on PD-L1 expression [134]. Based on recent single-cell sequencing (scSeq) data, paclitaxel may decrease critical antitumor immune cells in the TME while it may increase immunosuppressive macrophages, which could impact the efficacy of atezolizumab [135]. However, additional investigations are warranted to validate these findings. In addition to concurrent chemotherapy, another novel approach for the immuno-chemotherapy combination is to induce low-dose chemotherapy before immunotherapy. Patients with mTNBC received no induction or 2 weeks of low-dose cyclophosphamide, cisplatin, doxorubicin, and hypo-fractionated irradiation, followed by nivolumab, in the phase-II TONIC trial. The results suggested that doxorubicin or cisplatin induction, even for a short period, can shift the TIME toward an inflammatory state and enhance the response to nivolumab in TNBC [136]. Pembrolizumab and chemotherapy combination has consistently increased pCR in multiple clinical trials (I-SPY2 [137], KEYNOTE-173 [66], and KEYNOTE-522 [66]) in patients with early-stage TNBC. In July 2021, the FDA approved pembrolizumab as a neoadjuvant treatment for early-stage, high-risk TNBC alongside chemotherapy based on the improved pCR rates noted in these trials [138].

In the phase-II GeparNuevo trial, durvalumab or placebo was administered every 4 weeks in addition to chemotherapy and their efficacy was assessed. Addition of durvalumab significantly increased the pCR rates in the window cohort but not in the overall study population [139]. In the NeoTRIP trial, a separate trial of the PD-L1 inhibitor atezolizumab, the efficacy of eight cycles of carboplatin and nab-paclitaxel with or without atezolizumab was examined in high-risk TNBC. As an adjuvant treatment, four cycles of anthracycline regimen chemotherapy were given. The pCR rate in the ITT group was not significantly different with the addition of atezolizumab in a neoadjuvant setting [140]. Conversely, in the IMpassion031 study, atezolizumab given as a single drug with a standard chemotherapy regimen including doxorubicin and paclitaxel significantly increased pCR from 41% in the chemotherapy-alone arm to 58% in the ateza-plus-chemotherapy arm [141]. The IMpassion130 phase-3 trial, reported in 2018, showed that first-line treatment with atezolizumab plus nab-paclitaxel significantly improved PFS compared to placebo plus nab-paclitaxel in mTNBC. While the overall survival boundary was not crossed in this interim analysis and was not formally tested for statistical significance, numerical increases in median OS were observed in both the intention-to-treat and PD-L1-positive subgroups [17]. However, the FDA approval of atezolizumab was withdrawn due to the negative results from the IMpassion131 trial [134]. It should also be noted that the IMpassion130 and IMpassion131 trials used the SP142 assay for PD-L1 expression, which differs from other assays in detecting PD-L1 levels.

### 3.3. Combination of Immunotherapy with Radiotherapy

Radiotherapy (RT) is still a major treatment modality in TNBC even after recent advancements in endocrine therapy, chemotherapy, and targeted therapy for BC [112]. Numerous randomized trials have demonstrated that adjuvant radiation therapy decreases locoregional recurrence and improves survival in women with both early-stage and advanced-stage breast cancer. The impact of radiotherapy on immune signaling is still being elucidated [142,143]. Immune cells are attracted to the TME by radiation in several ways. It triggers the release of warning signals from dying tumor cells. DCs consume antigens from cancerous cells and deliver them to lymph nodes. The T cells are then exposed to them, which activates CD8+ and CD4+ T cells. Consequently, chemokines drive effector T-cell recruitment to tumors [144]. In BC patients, RT plus immunotherapy may produce systemic antitumor effects, especially when RT is administered at higher doses using more advanced techniques. Potential explanations for these systemic effects include the growth and dissemination of effector immune cells to distant sites because of the local immune priming by RT [145].

The adaptive phase-2 TONIC study, however, demonstrated a limited increase (~10%) in ORR with low-dose radiation combination as compared to nivolumab alone [136]. The best ORRs were observed with nivolumab and chemotherapy combinations (doxorubicin 35%, cisplatin ORR 23%) in this trial as mentioned above. A multicenter phase-2 trial assessed the safety and efficacy of pembrolizumab plus RT in patients with mTNBC (NCT02730130) [146]. This study discovered that the unselected PD-L1 population’s ORR in the ITT cohort was 17.6%, which was greater than the ORR of mTNBC patients who had previously received ICI monotherapy. Fifty patients who had received fewer than two lines of systemic therapy were included in a phase-II AZTEC trial to receive atezolizumab plus RT (NCT03464942) [147]. Patients were randomized to receive either 24 Gy stereotactic ablative radiotherapy (SABR) in three fractions or 20 Gy SABR in one fraction. Five days following the last RT segment, atezolizumab was initiated. There was no discernible variation in median progression-free survival (mPFS) between the two cohorts. TIL levels of 5% and the PD-L1 expression had little impact on the efficacy [147]. It would be interesting to see the results of numerous trials that are currently investigating the clinical benefit of RT in combination with ICI in women with TNBC.

### 3.4. Dual Antibody Combinations and Dual Immunotherapies

ADCs can interact with anti-PD-(L)1 agents to improve tumor control [148]. Sacituzumab govitecan targets TROP2 and delivers topoisomerase I inhibitor SN38 to the tumor. The clinical efficiency of this agent with pembrolizumab as a first-line treatment for mTNBC is currently being assessed [21]. Similarly, the safety and efficacy of ladiratuzumab vedotin an ADC that targets LIV-1 [149] is being evaluated in combination with pembrolizumab in a phase-Ib/II trial [150].

VEGF is a crucial factor in vascular endothelial cells that promotes angiogenesis, cell invasion, migration, proliferation, and survival, and enhances vascular permeability [151]. The combination of low-dose VEGFR inhibitor apatinib with anti-PD-1 agent camrelizumab and PARP1/2i fuzoloparib demonstrated a manageable safety profile and preliminary antitumor activity in patients with advanced TNBC [152]. The FUTURE-C-Plus trial demonstrated that CD8+ and/or PD-L1-positive patients benefit more from the combination of famitinib (VEGFRi), camrelizumab (ICI), and chemotherapy (nab-paclitaxel) [153]. The trial validated the safety, efficacy, and feasibility of triple therapy in TNBC and identified CD8+ positivity as a marker of favorable response with the triple therapy combination in the clinical setting [153].

Dual ICI therapies have been designed to overcome PD-(L)1 inhibitor resistance and reverse the tumor immunosuppressive microenvironment [154]. Durvalumab plus tremelimumab showed preliminary effectiveness and a manageable safety profile (NCT02536794) only in mTNBC patients (ORR 43%) while there was no response in ER+ BC [155]. Responders had higher expression of CD8, granzyme A, and perforin-1 post-therapy as compared to non-responders [155]. In the I/II SYNERGY (NCT03616886) trial, locally advanced or mTNBC patients were treated with a combination of oleclumab (anti-CD73 antibody), durvalumab, and chemotherapy (carboplatin and paclitaxel). The phase-II part of this trial was randomized 1:1 with/without oleclumab. However, this trial did not meet its primary endpoint (insignificant clinical benefit at 24 weeks) [156].

Similarly, in patients with mTNBC, a phase-I trial (NCT03256344) assessed the safety of intrahepatic injection of T-VEC in combination with intravenous atezolizumab [157]. The five TNBC patients in their DLT cohorts did not have any dose-limiting toxicities (DLT); however, the majority of TNBC patients (90%) in this trial presented with grade 3 adverse events (AEs) with limited evidence of antitumor activity [157]. A first-in-human trial that included multimodality treatments including chemoradiation, NK cell therapy, typhoid conjugate vaccine (TCV), and a PD-L1 inhibitor as third-line therapy for mTNBC (NCT03387085) found the combination treatment to be safe and well tolerated, and it achieved a 56% ORR in early efficacy results [158]. These encouraging results suggest that multimodal treatment will be the way forward for the management of recurrent and metastatic TNBC and may lead to the development of additional multimodality clinical trials.

## 4. Conclusions and Future Perspectives

In conclusion, while immunotherapy has made significant strides in the treatment of TNBC, there is still much progress to be made in optimizing these approaches for broader clinical benefit. The successes seen with ICIs have paved the way for more innovative strategies, yet challenges such as limited response rates and the development of resistance must be addressed. Moving forward, the integration of immunotherapy into “window of opportunity” trials could provide crucial insights into the tumor immune microenvironment and help identify biomarkers that predict responses to treatment. Furthermore, combination therapies hold considerable promise in enhancing the efficacy of immunotherapies in TNBC. Combining immune checkpoint inhibitors with chemotherapy, targeted therapies, radiation, or other immunomodulatory agents such as cytokines and oncolytic viruses may enhance the immune response and overcome resistance mechanisms. The use of adoptive cell therapies, such as CAR-T cells and TILs, may also be explored in combination with immune checkpoint inhibitors to boost the antitumor immune response. Additionally, the future of immunotherapy in TNBC will likely involve more personalized approaches, tailoring treatments based on individual tumor profiles and immune characteristics. Identifying new biomarkers, improving patient selection, and understanding the mechanisms of resistance will be critical in advancing the field. Continued exploration of novel immunotherapeutic strategies, such as cancer vaccines and bispecific antibodies, may further expand the treatment arsenal for TNBC. While immunotherapy has shown promise in TNBC, its full potential will likely be realized through the strategic design of combination therapies and personalized treatment approaches informed by ongoing research.

The advancements in immunotherapy have brought renewed optimism for women battling local, recurrent, and metastatic TNBC. Despite the success of integrating immunogenic chemotherapy with ICIs, particularly in early-stage TNBC, trials evaluating adjuvant immune checkpoint inhibitors in operable TNBC like IMpassion030 (NCT03498716) have yielded unsatisfactory outcomes. These innovative treatments provide additional therapeutic options for patients who have undergone extensive prior therapies and developed resistance. Over the past few years, a growing number of small-molecule inhibitors including tyrosine kinase inhibitors (EGFR and VEGFR), serine/threonine kinase inhibitors (ATM, ATR, AKT, CDK1, CDK4/6, CHK1, DNA-PKcs, mTOR, PI3K, and WEE1), dual specific kinase inhibitors (TTK1, MEK), and proteasome, PARP, and epigenetic (HDAC) inhibitors have been tested in TNBC as monotherapy or in combination with other targeted agents or ICI [159,160]. While several of these therapeutic combinations are still in the experimental stage or undergoing clinical trials, they represent promising avenues for the treatment of TNBC (Table 3). Despite the clinical success of targeted small-molecule treatments for TNBC, drug resistance is still an ongoing challenge. Other possibilities include combination treatments, novel mutation inhibitors, and multi-targeted drugs. Novel therapeutic targets, such as BUB1, LIG4, Hh, and XPO1, are being investigated in preclinical or clinical studies for targeting TNBC [160,161,162,163,164]. It is anticipated that in the near future, these small-molecule inhibitors and immunotherapy will be able to work together to increase the antitumor activity of these drugs. Combination therapy may prove effective, but uncertainties still exist relating to method, sequence, dosage, and duration with a careful eye toward balancing toxicity and affordability.

Integrating artificial intelligence (AI) in treatment planning may revolutionize the prediction of therapeutic outcomes, enabling personalized precision medicine. High-throughput sequencing and AI can identify novel molecular markers and gene signatures, aiding clinicians in selecting the most effective patient-specific therapies. This may then assist the clinical care team in optimally selecting a “tailored patient-specific” therapy that is most likely to work, thus opening new horizons in precision medicine [165]. This will require scientists, organizations, and medical professionals to work together to develop databases, remove technological obstacles, and support the creation of AI-assisted systems that can precisely identify the target populations/patients, predict the efficacy and prognosis, and strongly support the use of AI-assisted treatment. We anticipate that with the development of novel drug discovery/prediction datasets, large-scale genomic/genetic datasets, and immune signatures combined with large multicenter clinical trials will soon make significant strides in treating TNBC efficiently [166,167,168,169,170].

## Figures and Tables

**Figure 1 cancers-16-03250-f001:**
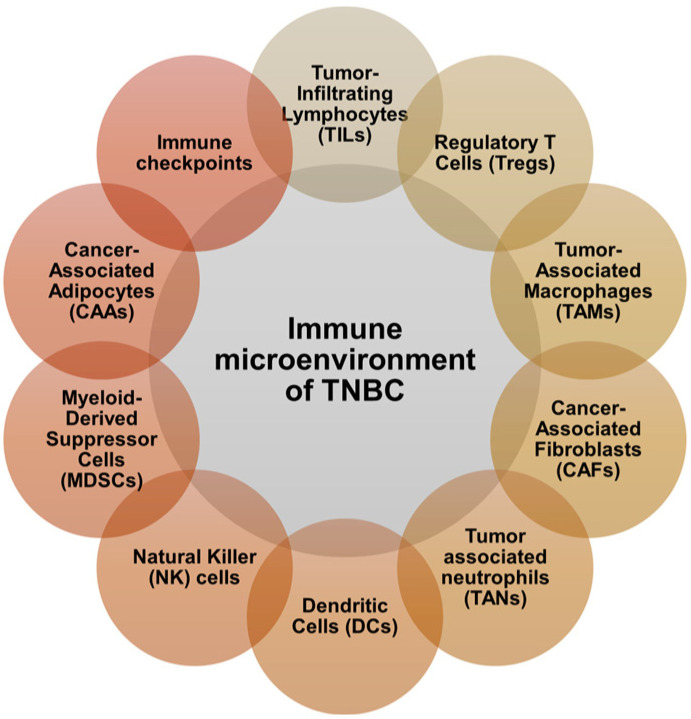
Immunological components of TNBC tumor microenvironment. This illustration sets forth the immunological components of the TNBC tumor microenvironment, which includes tumor-infiltrating lymphocytes (TILs), regulatory T cells (Tregs), tumor-associated macrophages (TAMs), cancer-associated fibroblasts (CAFs), tumor-associated neutrophils (TANs), dendritic cells (DCs), natural killer (NK) cells, myeloid-derived suppressor cells (MDSCs), cancer-associated adipocytes (CAAs), and immune checkpoints.

**Figure 2 cancers-16-03250-f002:**
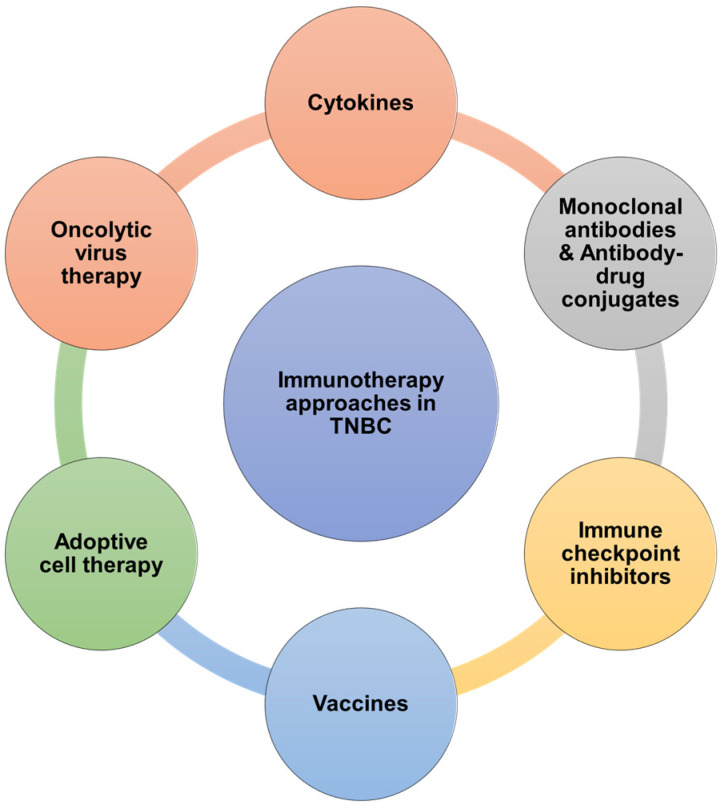
Current clinical immunotherapy approaches in TNBC. Clinical immunotherapy approaches for TNBC have been diversified in recent years and this illustration summarizes those strategies which include cytokines, mAbs/ADCs, immune checkpoint inhibitors, vaccines, adoptive cell therapies, and oncolytic virus therapy.

**Figure 3 cancers-16-03250-f003:**
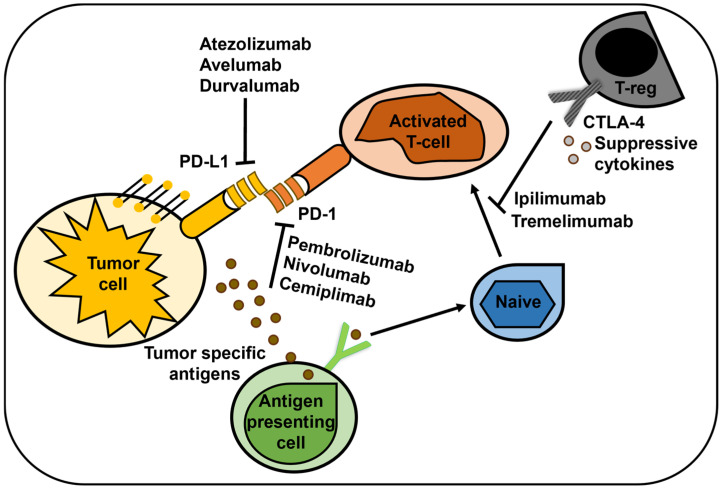
Mechanism of action of ICIs targeting PD-L1, PD-1, and CTLA-4. T-cell inactivation and the prevention of tumor cell death are caused by their binding to the corresponding ligands on the surface of cancer cells. Immune checkpoint inhibition promotes anticancer activity and promotes T-cell activation.

**Table 1 cancers-16-03250-t001:** Updated list of FDA-approved drugs to treat TNBC (accessed information from https://www.fda.gov/ on 26 July 2024).

Drug Class	Agents
Cytotoxic Chemotherapy	Carboplatin, Docetaxel, Doxorubicin, Epirubicin, Ixabepilone, Liposomal doxorubicin, Nab-paclitaxel, Paclitaxel, Vinorelbine, Cisplatin
Immunotherapy	Atezolizumab, Pembrolizumab
Antibody–Drug Conjugates	Sacituzumab govitecan, Trastuzumab deruxtecan (TNBC with low/ultra-low HER2 expression)
PARP Inhibitors	Olaparib, Talazoparib

**Table 2 cancers-16-03250-t002:** Combination of immunotherapy with other treatment modalities evaluated in clinical trials for TNBC (accessed information from https://www.clinicaltrials.gov/ on 17 June 2024).

Target	Interventions	Clinical Status and Identifier	Status
PARP and PD-1	Drug: Niraparib Biological: Pembrolizumab	Phase I/II NCT02657889	Completed
PARP and PD-L1	Drug: Avelumab Phase 1b Drug: Talazoparib Phase 1b Drug: Avelumab Phase 2 Drug: Talazoparib Phase 2	Phase Ib/II NCT03330405	Completed
PD-1	Biological: Pembrolizumab Drug: Nab-paclitaxel Drug: Paclitaxel Drug: Gemcitabine Drug: Carboplatin Drug: Normal Saline Solution	Phase III NCT02819518	Completed
PD-1	Drug: Eribulin Mesylate Drug: Pembrolizumab	Phase Ib/II NCT02513472	Completed
PD-L1	Drug: Atezolizumab (MPDL3280A) Drug: Nab-Paclitaxel Drug: Placebo	Phase III NCT02425891	Completed
PD-L1	Drug: Atezolizumab Drug: Nab-paclitaxel	Phase Ib NCT01633970	Completed
PD-L1	Drug: Atezolizumab (MPDL3280A) Drug: Atezolizumab Placebo Drug: Paclitaxel	Phase III NCT03125902	Completed
PD-1	Drug: Nivolumab Radiation: Radiation therapy Drug: Low-dose doxorubicin Drug: Cyclophosphamide Drug: Cisplatin	Phase II NCT02499367	Ongoing
PD-1	Biological: Pembrolizumab Drug: Nab-paclitaxel Drug: Anthracycline (doxorubicin) Drug: Cyclophosphamide Drug: Carboplatin Drug: Paclitaxel	Phase I NCT02622074	Completed
PD-1	Biological: Pembrolizumab Drug: Carboplatin Drug: Paclitaxel Drug: Doxorubicin Drug: Epirubicin Drug: Cyclophosphamide Drug: Placebo Biological: GM-CSF	Phase III NCT03036488	Ongoing
PD-L1	Drug: MEDI4736 (Durvalumab) Drug: Placebo Drug: Nab-Paclitaxel Drug: Epirubicin Drug: Cyclophosphamide	Phase II NCT02685059	Completed
PD-L1	Drug: Carboplatin Drug: Abraxane Drug: MPDL3280A (Atezolizumab) Procedure: Surgery Drug: Anthra	Phase III NCT02620280	Ongoing
PD-L1	Drug: Atezolizumab (MPDL3280A) Drug: Placebo Drug: Nab-paclitaxel Drug: Doxorubicin Drug: Cyclophosphamide Drug: Filgrastim Drug: Pegfilgrastim	Phase III NCT03197935	Completed
PD-1	Drug: Pembrolizumab Radiation: Radiotherapy	Phase II NCT02730130	Completed
PD-L1	Radiation: SABR Drug: Atezolizumab	Phase II NCT03464942	Completed
PD-1 and LIV-1	Drug: Ladiratuzumab vedotin Drug: Pembrolizumab	Phase Ib/II NCT03310957	Ongoing
PD-L1 and AKT	Drug: Atezolizumab Drug: Ipatasertib Drug: Paclitaxel Drug: Placebo for Atezolizumab Drug: Placebo for Ipatasertib	Phase III NCT04177108	Completed
PD-1, PARP, and VEGFR-2	Drug: SHR-1210 + Apatinib + Fluzoparib	Phase I NCT03945604	Completed
PD-1, VEGFR-2, c-KIT, and PDGFRb	Drug: Camrelizumab + nab-paclitaxel + famitinib	Phase II NCT04129996	Completed
PD-L1 and CD73	Drug: Paclitaxel Drug: Carboplatin Drug: MEDI4736 Drug: MEDI9447	Phase I/II NCT03616886	Ongoing
PD-L1 and modified oncolytic herpes virus	Biological: Talimogene Laherparepvec Biological: Atezolizumab	Phase Ib NCT03256344	Ongoing
PD-L1	Avelumab, SBRT, haNK, and 15 other interventions/treatments	Phase I/II NCT03387085	Completed

**Table 3 cancers-16-03250-t003:** List of non-immune therapies under clinical trials for TNBC treatment (accessed information from https://www.clinicaltrials.gov/ on 17 June 2024).

Target	Interventions	Clinical Status and Identifier	Status
EGFR	Drug: Metformin Drug: Erlotinib	Phase I NCT01650506	Completed
PI3K	Drug: BKM120	Phase II NCT01790932	Completed
PI3K	Drug: BKM120 and Olaparib Drug: BYL719 and Olaparib	Phase I NCT01623349	Completed
PI3K	Drug: BYl719	Phase II NCT02506556	Completed
AKT	Drug: Ipatasertib Drug: Paclitaxel Drug: Placebo	Phase II NCT02301988	Completed
AKT	Drug: Ipatasertib Drug: Paclitaxel Drug: Placebo	Phase II NCT02162719	Completed
AKT	Drug: Paclitaxel Drug: AZD5363 Drug: Placebo	Phase II NCT02423603	Unknown
AKT	Drug: Capivasertib Drug: Paclitaxel Drug: Placebo	Phase III NCT03997123	Ongoing
AKT	Drug: Capivasertib Other: Laboratory Biomarker Analysis Drug: Olaparib Other: Pharmacological Study Drug: Vistusertib	Phase Ib NCT02208375	Ongoing
AKT	Drug: GSK1120212 Drug: GSK2141795	Phase I NCT01138085	Completed
mTOR	Drug: Doxil Drug: Bevacizumab Drug: Temsirolimus	Phase I NCT00761644	Completed
mTOR	Drug: Everolimus	Phase II NCT01931163	Completed
mTOR	Drug: Everolimus Drug: Eribulin mesylate Other: Pharmacological study Other: Laboratory biomarker analysis	Phase I NCT02120469	Completed
mTOR	Drug: Everolimus Drug: Eribulin	Phase I NCT02616848	Completed
CDK4/6	Drug: Trilaciclib Drug: Gemcitabine Drug: Carboplatin	Phase 2 NCT02978716	Completed
CDK4/6	Drug: Trilaciclib Drug: Gemcitabine Drug: Carboplatin	Phase 2 NCT02978716	Completed
ATR	Drug: M6620 Drug: Gemcitabine Drug: Cisplatin Drug: Etoposide Drug: Carboplatin Drug: Irinotecan	Phase I NCT02157792	Completed
ATR	Drug: Olaparib Drug: Ceralasertib Drug: Adavosertib	Phase 2 NCT03330847	Ongoing
ATR	Procedure: Biopsy Drug: Capivasertib Drug: Ceralasertib Biological: Durvalumab Drug: Olaparib Other: Quality-of-Life Assessment Drug: Selumetinib	Phase II NCT03801369	Ongoing
CHK1	Drug: LY2606368	Phase II NCT02203513	Completed
WEE1	Drug: Cisplatin Drug: AZD1775	Phase II NCT03012477	Completed
MEK	Drug: GSK1120212 Drug: GSK2141795	Phase I NCT01138085	Completed
MEK	Drug: Akt Inhibitor GSK2141795 Other: Laboratory Biomarker Analysis Drug: Trametinib	Phase II NCT01964924	Completed
MEK	Drug: Ipatasertib Drug: Cobimetinib	Phase I NCT01562275	Completed
MET, VEGFR2, RET, AXL, FTL3, etc.	Drug: Cabozantinib	Phase II NCT01738438	Completed
VEGF, PDGFR, HGF, etc.	Drug: Paclitaxel Drug: Carboplatin Drug: Sunitinib	Phase I/II NCT00887575	Completed
VEGF, PDGFR, HGF, etc.	Drug: SU011248 Drug: Chemotherapy	Phase II NCT00246571	Completed
Aurora-A, VEGFR, FGFR	Drug: ENMD-2076	Phase II NCT01639248	Completed
EGFR, HER2	Drug: Veliparib + Lapatinib	Phase: N/A NCT02158507	Ongoing
PI3K, mTOR	Drug: Prexasertib Drug: Cisplatin Drug: Cetuximab Drug: G-CSF Drug: Pemetrexed Drug: Fluorouracil Drug: LY3023414 Drug: Leucovorin	Phase I NCT02124148	Completed
PARP	Drug: Pamiparib	Phase I/II NCT03333915	Completed
PARP	Drug: Talazoparib	Phase II NCT03499353	Completed
PARP	Drug: Olaparib	Phase II NCT02681562	Completed
PARP	Drug: Olaparib Radiation: Radiation therapy	Phase I NCT03109080	Completed
PARP	Drug: Iniparib Drug: Gemcitabine Drug: Carboplatin	Phase II NCT01045304	Completed
PARP	Drug: Cyclophosphamide Drug: Placebo Drug: Doxorubicin Drug: Paclitaxel Drug: Carboplatin Drug: Veliparib Drug: Placebo	Phase III NCT02032277	Completed
HDAC	Drug: Chidamide + Cisplatin	Phase II NCT04192903	Completed
HDAC	Drug: Entinostat	Phase I NCT03361800	Terminated
HDAC	Drug: Romidepsin Drug: Cisplatin Drug: Nivolumab	Phase I/II NCT02393794	Ongoing
SMO	Drug: LDE225 Drug: Docetaxel	Phase I NCT02027376	Completed
XPO1	Drug: Selinexor	Phase II NCT02402764	Completed

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
