# Peer review of "Present and Future of Immunotherapy for Triple-Negative Breast Cancer"

_cancers, 2024, doi:10.3390/cancers16193250_

Round 1
Reviewer 1 Report
Comments and Suggestions for Authors
In this review, Sriramulu et al. reviewed the present and possibly future of immunotherapy in Triple-negative breast cancer. This review however, major issues:
1) it is very long
2) It is not organized correctly. the long definition of different types of cells in the tumor are not necessarily. They used the general term as "Current approaches", should have been currently approved agents. Alternatively, within every approach, it could have been divided into approved agent and agents under investigation.
3) The Tables are not organized. There are no dividers between each trial, so it is not possible to link the intervention with the trial number sometimes. Also, because they are not organized, the intervention is repeated multiple times just because it is written this way on the NCT.gov website. This, however, is not appropriate for a review article.
Author Response
Reviewer 1:
We thank the reviewer #1 for their thorough evaluation of our manuscript and helpful suggestions that resulted in significant improvement.
In this review, Sriramulu et al. reviewed the present and possibly future of immunotherapy in Triple-negative breast cancer. This review, however, major issues:
- it is very long.
We acknowledge that the content may appear extensive; however, our goal was to provide readers with detailed information in this updated article. However, as suggested by the reviewer, we have removed the section titled “Immune Microenvironment of TNBC” (formerly Section 2) that detailed the components of the immune tumor microenvironment. Instead, we have incorporated the key information into the Introduction section.
- It is not organized correctly. the long definition of different types of cells in the tumor is not necessary. They used the general term "Current approaches", which should have been currently approved, agents. Alternatively, every approach, it could have been divided into approved agents and agents under investigation.
As addressed in comment 1, we have removed the section discussing the different cell types in the tumor microenvironment. Additionally, we have revised the title in accordance with the reviewer’s suggestion. For clarity, we have underlined the approved agents. Please refer to page 3, line 107.
- The Tables are not organized. There are no dividers between each trial, so it is not possible to link the intervention with the trial number sometimes. Also, because they are not organized, the intervention is repeated multiple times just because it is written this way on the NCT.gov website. This, however, is not appropriate for a review article.
We apologize for the lack of clarity in the previous version. We have now reorganized the tables, adding dividers between each trial, and ensured that the interventions are not repeated multiple times. Our goal is to present this information in the table to help clinicians better understand the recent updates in the field. Please refer to page 3, line 104, page 10, line 392, page 17, line 635.
Reviewer 2 Report
Comments and Suggestions for Authors
This is a review on the role (present and future) of immunotherapy in triple negative breast cancer. The topic is of great interest and clinically relevant at the moment. The review is comprehensive and very detailed, provides many information on drugs in development, but fails to provide a clear overview of the present state of art and future of the field. Authors may revise the text to underline in any paragraph currently approved and used drugs at first (if any) and then adding details on drug in development.
Moreover, "conclusions and future perspectives" paragraph needs extensive revisions and expansion to really convey authors perspective of the future of immunotherapy (use in window of opportunity trials? combine with?). In addition, in this section authors state that combination of chemotherapy+ICI is particularly effective in late stage and metastatic TNBC and is an option for patients who have undergone extensive prior therapies and developed resistance. However, this is not the case of immunotherapy that proved its greatest value in the early (neoadjuvant setting or in first line of treatment in the metastatic setting).
Some other points to be addressed:
- PARP inhibitors were approved by the FDA. (pag 1,2 lines 44-45) should be moved in another section/paragraph
- Tables should be formatted and revised to become more easily understandable. I would suggest to remove drugs approved for other BC subtypes and mantain only drugs for TNBC in table 1
- The immunogenicity of TNBC goes beyond BRCA1/2 mutations (these are present in a minority of the cases) Please check and rephrase lines 86-91
- In paragraph 3 please add a brief introduction and please refer to figure 2 before introducing paragraph 3.1
Comments on the Quality of English LanguageExtensive English language revision is needed to make the work suitable for publication: adverbs are frequently used unappropriately. Many sentences need to be rephrased. Please check also punctuation, that is frequently uncorrect.
Some sentences below just as an example:
- “respond better” (line 52)
- “This allows for a decrease in the tumor burden and an in vivo assessment of treatment response, such that pathological complete response (also known as pCR) is a useful prognostic marker for survival” (line 56-58)
- Similarly, patients do not tolerate systemic treatment of recombinant IFN-α as well (lines 198-199)
- However, olaparib had minimal impact on the overall patient-reported quality of life (line469)
Author Response
Reviewer 2:
We thank the reviewer #2 for their extensive feedback on our work.
This is a review of the role (present and future) of immunotherapy in triple-negative breast cancer. The topic is of great interest and clinically relevant at the moment. The review is comprehensive and very detailed and provides many information on drugs in development, but fails to provide a clear overview of the present state of the art and future of the field. Authors may revise the text to underline in any paragraph currently approved and used drugs at first (if any) and then add details on drugs in development.
We appreciate Reviewer #2's feedback on our work. Following the reviewer's suggestions, we have revised the text and underlined the currently approved drugs. We hope this revision provides clearer information for readers.
Moreover, the "conclusions and future perspectives" paragraph needs extensive revisions and expansion to convey the author's perspective on the future of immunotherapy (use in window of opportunity trials? combine with?). In addition, in this section authors state that a combination of chemotherapy + ICI is particularly effective in late-stage and metastatic TNBC and is an option for patients who have undergone extensive prior therapies and developed resistance. However, this is not the case of immunotherapy which proved its greatest value in the early (neoadjuvant setting or in the first line of treatment in the metastatic setting).
Thank you for pointing out the error. We apologize for the incorrect statement. We have revised the statement and expanded the "Conclusion and Future Perspectives" section to reflect our perspective. Please refer to page 16, lines 591 to 615.
Some other points to be addressed:
- PARP inhibitors were approved by the FDA. (page 1,2 lines 44-45) should be moved in another section/paragraph
We have removed the sentence and relocated it to the appropriate section. Please refer to page 12, lines 398-399.
- Tables should be formatted and revised to become more easily understandable. I would suggest removing drugs approved for other BC subtypes and maintaining only drugs for TNBC in Table 1
As suggested by the reviewer, we have reformatted and revised the table to improve clarity for readers. We have removed information on other breast cancer subtypes and included only the drugs approved specifically for TNBC. Please refer to page 3, line 104.
- The immunogenicity of TNBC goes beyond BRCA1/2 mutations (these are present in a minority of the cases) Please check and rephrase lines 86-91
We have revised those lines as suggested. Please refer to page 2, lines 70-78 with appropriate references.
- In paragraph 3 please add a brief introduction and please refer to figure 2 before introducing paragraph 3.1
Thank you for your feedback. We have now added a brief introduction to the section and referenced Figure 2. Please see page 3, lines 109-116.
Extensive English language revision is needed to make the work suitable for publication: adverbs are frequently used inappropriately. Many sentences need to be rephrased. Please check also punctuation, which is frequently incorrect.
Some sentences below just as an example:
- “respond better” (line 52)
- “This allows for a decrease in the tumor burden and an in vivo assessment of treatment response, such that pathological complete response (also known as pCR) is a useful prognostic marker for survival” (lines 56-58)
- Similarly, patients do not tolerate systemic treatment of recombinant IFN-α as well (lines 198-199)
- However, olaparib had minimal impact on the overall patient-reported quality of life (line469)
As suggested by the reviewer, we have addressed the English language revision by using appropriate grammatical software and having it reviewed by a native English-speaking colleague.
Round 2
Reviewer 1 Report
Comments and Suggestions for Authors
The review has improved. However, the FDA-approved agents in Table 1 currently include Pembrolizumab and Atezolizumab as targeted therapy. They should be better described as immunotherapy, especially given the focus of this review on immunotherapy. Targeted therapy usually targets cancer cells rather than immune cells.
In addition, the PARP inhibitors Olaparib and Talazoparib are approved for treating TNBC with BRCA1/2 germline mutations (more common in TNBC than any other subtype). This should be included in Table 1 under PARP inhibitors.
Author Response
Reviewer 1:
We thank reviewer #1 for their extensive feedback on our work.
- The review has improved. However, the FDA-approved agents in Table 1 currently include Pembrolizumab and Atezolizumab as targeted therapy. They should be better described as immunotherapy, especially given the focus of this review on immunotherapy. Targeted therapy usually targets cancer cells rather than immune cells.
Per the reviewer’s suggestion, we have replaced “targeted therapy” with “immunotherapy.” Please see page 3, line 105 of Table 1 for the update.
- In addition, the PARP inhibitors Olaparib and Talazoparib are approved for treating TNBC with BRCA1/2 germline mutations (more common in TNBC than any other subtype). This should be included in Table 1 under PARP inhibitors.
PARP inhibitors have now been added to Table 1.
Reviewer 2 Report
Comments and Suggestions for Authors
The authors Made significant revision of the english language and overall quality of the manuscript has improved thanks to the review made. Just 2 minor editing:
- niraparib is approved for the treatment of gynaecologic cancer but not breast cancer. Please make it clear in the text
- in the conclusion proteasome inhibitors are called PARP
Comments on the Quality of English LanguageExtensive revision has been made. Now the work a suitable for publication
Author Response
Reviewer 2:
We thank reviewer #2 for their thorough evaluation of our manuscript and helpful suggestions, which resulted in significant improvement.
- The authors made significant revisions of the English language and the overall quality of the manuscript has improved thanks to the review made. Just 2 minor editing:
- niraparib is approved for the treatment of gynecologic cancer but not breast cancer. Please make it clear in the text
Thank you for pointing out the error. We apologize for the incorrect statement. We have now revised the statement. Please refer to page 12, line 398 for the revision.
- In conclusion, proteasome inhibitors are called PARP.
Thank you for your feedback. We have revised the text. Please see page 16, line 623 for reference.